# Security Enhancement in Smart Logistics with Blockchain Technology: A Home Delivery Use Case

Tirajet Chukleang [1] and Chanankorn Jandaeng [2,*]

1   Management of Information Technology, School of Informatics, Walailak University,
    Nakhon Si Thammarat 80160, Thailand
2   Informatic Innovation Center of Excellence (IICE), School of Informatics, Walailak University,
    Nakhon Si Thammarat 80160, Thailand
*   Correspondence: chatchanan.ja@mail.wu.ac.th

**Abstract:** Home delivery (B2C) experienced rapid growth during the COVID-19 pandemic, especially food delivery. Nonrepudiation is a problem in security and affects delivery. Blockchain technology is a new technology that addresses this issue. This paper proposes preventing nonrepudiation in home delivery through blockchain technology. We also design a data structure and smart contract for distributed application. In addition, we evaluate the performance of the proposed framework in terms of transaction fees and response times. We experimented on the blockchain emulator, stored data in RDBMS, and implemented a client with a mobile application. The data in the smart contract were directly impacted by the gas used and the response time. Primarily, the function processes the large data size and involves high transaction fees and long response times. The gas used accounts for 9061 times of data length, while the response time accounts for 2.84 times of data length. Finally, we propose a security policy for the proposed framework.

**Keywords:** blockchain application; home delivery; smart contract; logistics

## 1. Introductions

Industry 4.0 is primarily concerned with a technology-driven industrial paradigm based on two fundamental concepts: The Internet of Things and cyber-physical systems (CPS). The high speed of the Internet is the key to improving the Industrial Internet of Things (IIoT), while the level of maturity of an Industry 4.0 system is shown by how well automation and intelligence work together in a CPS. As the primary function of a company or a supply chain, logistics has been significantly affected by recent technological advancements and innovation. This process is the core of services/products/goods distribution from producer to customer in Industry 4.0, called smart logistics [1]. Modern technology, such as the IoT, 5G communication, sensors, RFID, smart product, actuators, and intelligent machines, characterizes the new paradigm. Consequently, a large amount of data will be generated, and new challenges will emerge.

Security and privacy preservation are essential concerns for Industry 4.0 applications. There may be chances of unauthorized data breaching or information leakage, leading to financial losses to Industry 4.0-based applications. Because of the complexity and lack of transparency of traditional smart logistics, it is of great interest for the stakeholders involved in the logistics process to introduce and develop blockchain technology to enhance the logistics processes in the supply chain, making them more sustainable [2].

In addition, the main security requirements of this problem are reliability, nonrepudiation, and traceability. In order to eliminate fraud by the platform owner, a decentralized system or distribution system is one possible solution. The traditional technique is cryptographic to increase reliability, whereas a digital signature with asymmetric keys supports nonrepudiation and traceability. However, these methods are centralized systems. The fraud by the platform owner still occurs. Thus, blockchain technology is a solution to the problem.

Blockchain is a decentralized system that is transparent and secures services. The blockchain system spreads the transactions out instead of keeping the ledger data in one place so that each person can verify their transactions. So, there are fewer chances that a transaction will be changed. It is also not possible if the network has many nodes. In addition, the characteristics of a blockchain are reliability, traceability, and authenticity (nonrepudiation) of the information that secures the transaction [1].

For the logistics domain, blockchain technology was introduced to improve the ability, security, and traceability. In addition, blockchain is expected to impact the logistics industry considerably [3], including making exchanges between different channels easier and encouraging the development of laws and regulations [1]. Roberto et al. proposed utilizing smart contracts to eliminate intermediaries and simplify logistical operations. In addition, a multiagent system was implemented to organize the complete logistics service and smart contracts and their terms. They integrated the new model, smart contracts, and a multiagent system to improve the present logistics system by significantly improving the organization, security, and delivery times [4]. Hackius et al. presented a scheme that relies on smart contracts to eliminate intermediates and improve logistics. Furthermore, the multiagent system handles logistics, smart contracts, and compliance. They use the smart contract with a multiagent system to improve the logistics system's security while reducing delivery times [5]. In the dimension of service quality, Chen et al. proposed a platform for the quality management of perishable supply chain logistics named EBIQS. This platform considers the resource constraint devices. They proposed adaptive data smoothing and data compression. Given the massive amount of IoE data, for the ease of data block validation, generation, and storage in the edge blockchain, the recommended strategy is compression. Additionally, they made a synchronization engine that coordinates the fix and a mobile edge gateway to keep IoE sensing and data exchange going during transport. This engine makes sure that the sensing data are correct and can be seen in real time [6].

From the traceability perspective, Terzi et al. introduced the idea of smart contracts to ensure the logistics industry's security, integrity, and reliability. Then, they proved their idea by implementing it in real scenarios. First, they identified the food ingredients with the blockchain-based application. Then, they ensure that the items are successfully transferred from the manufacturer to the customer. The solution showed that the transport was transparent and authentic. IoT-Blockchain is a driven traceability technique for improved safety measures in the food supply chain. This technique records the status of food items and validates the information source. Blockchain makes it feasible for the logistics industry to realize data exchange and storage to ensure that data are available, traceable, and verifiable. Additionally, blockchain makes it immediately clear at any point on the network when unsafe food is identified, and access to it is blocked [7]. Helo et al. proposed a blockchain-based tracking system with RFID devices in authentication [8]. George et al. applied this interoperation to improve food safety and traceability [9].

Moreover, Wang et al. proposed a traceability system for the food industry based on blockchain and the Internet of Things. This approach stores the simple blockchain storage structure and all information regarding the product in an RFID tag. At the same time, an RFID reader with a blockchain light node is used to exchange data between RFID and the blockchain platform [10].

This research proposes a blockchain-based framework for smart business-to-customer (B2C) logistics (home delivery). The main objective is to securely trace the order and improve the nonrepudiation of transactions for home delivery services. The content comprises two parts: (1) the framework and its smart contract and (2) the performance evaluation, addressing transaction cost and response time. Our framework is a guideline for implementing distributed application based on blockchain technology for the business-to-customer logistics industry.

The main contributions of this research are:

- Framework design of the distributed application to prevent the nonrepudiation problem, tracing the activity.

- A smart contract for monitoring the B2C logistics.
- The performance analysis of the blockchain-based application in terms of transaction fee (gas use analysis) and response time.

This research is structured as follows. First, we briefly explain smart logistics and home delivery, blockchain technology, and the Ethereum network. After that, the food delivery and its security flaws are analyzed in Section 3. Our proposed framework is described in Section 4 and evaluated in Section 5. Finally, and we conclude our study in Section 6.

## 2. Background and Related Work

### 2.1. Smart Logistics and Home Delivery

This study takes parcel delivery as an example to support the proposed framework. The goods are picked up by the shipper and transported to a departure terminal. Following this, they are transported by line haul to the arrival terminal. Finally, local delivery sends the parcel to the customer at home. If the customer is not at home, the package is returned to the terminal and delivered the next day [11].

The importance of home delivery is especially pronounced in the conditions of impossible movement, which was confirmed during the COVID-19 pandemic. Shopping habits have changed to online shopping, especially food delivery from local restaurants. People are more likely to order food delivery via any online platform. According to Kasikorn Research Center, the food delivery business grew by 169.40% in 2020 compared to 2016. Thai people have been increasingly interested in joining the food delivery industry since the coronavirus spread. Previously, food delivery was limited to companies that sold products such as pizza and fried chicken, or privately owned business, etc. Some shops advertised their products via websites or social networks and delivered them via transportation services. However, these services were only popular among some groups of people.

The growth of food delivery services is also redistributing the revenue to a new stakeholder, a motorcycle food delivery driver, or a rider. It is estimated that the income distribution within the country accounts for at least 90% of the value of food delivery services to the customer. Today, the food delivery service industry has grown significantly and is ingrained in daily life. There are several food delivery platforms in Thailand. Users can access them through either web applications or mobile applications. Although each platform has a similar functionality, the differences are that the restaurants increase the service charge, advertising, and food costs.

### 2.2. Blockchain Technology

Nakamoto [12] proposed blockchain technology to create an electronic monetary system without an intermediary. Several information methods have emerged to address the mentioned issues. Blockchain technology is one answer. This approach encrypts the transaction within the encrypted block, connects it to other encrypted blocks, and regularly adds new blocks. The encryption of block data using the distributed consensus algorithm is an approach for maintaining data security.

Blockchain is a peer-to-peer network that uses the TCP protocol and has a random topology. Each node randomly peers at other nodes [13]. Decentralized transactions are also possible using blockchain technology. The blockchain system's method is more sophisticated in terms of technological components. When a client or application publishes a transaction to the blockchain network, miners validate it, and nodes verify transactions. Each miner node verifies the transaction's authenticity, guaranteeing that the sender has sufficient funds to cover the transaction's processing fee or sufficient funds to cover the transaction's processing fee in the case of a cash transfer system. It also double-checks the recipient's address and other relevant details. The miner node then generates a block containing the transactions that have been verified. However, the blockchain network will choose only one miner's block. The miner node is chosen using a consensus mechanism. The block is broadcast to all other nodes if any node is elected. In addition, the blockchain network and transaction verification fees reward it. The block is linked to the network

through a link or chain due to the hash function of the data in the block. The hash value will change if the blog is updated, revealing the inconsistency of the transaction.

Numerous blockchain platforms are available, including Ethereum, Stellar, IBM Blockchain, Hyperledger Fabric, Hyperledger Sawtooth, and R3 Corda. Hyperledger Fabric and Sawtooth are examples of open source blockchain frameworks. Hyperledger Fabric is a modular, general-purpose framework with special features for managing identities and controlling access. This makes it useful for a wide range of industrial applications. Hyperledger Sawtooth is the blockchain framework for businesses. It is a framework that makes it easy to build applications for networks and distributed ledgers. Both frameworks are used to set up a private blockchain. The R3 Corda is a distributor ledger company that was one of the first to use blockchain technology. Corda makes applications for the blockchain that help analyze and handle sensitive data from multiple parties. However, this platform is a commercial product. Stellar is an open-source protocol for using the Stellar Consensus Protocol to trade money or tokens. Thus, the Steller does not really work for our requirements. For this use case to work, the blockchain needs to be open-source and public. Fraud from the platform owner will be prevented by the public blockchain. Additionally, the Ethereum network is a well-known platform that many businesses use. To help with the implementation, there are toolchain and network blockchain emulators [14].

Blockchain technology effectively solves the pain points of traditional traceability systems by using its distributed storage, encryption algorithms, and timestamps [15]. Additionally, blockchain technology serves as a foundation for decentralized apps, land rights management [16], the prevention of government corruption [17], insurance [18], and the digital patent [19], which are also all examples of blockchain-based decentralized applications.

Moreover, blockchain technology prevents the nonrepudiation problem. The nonrepudiation problem is a key to the objectives of cybersecurity: confidentiality, integrity, availability, authenticity, authorization, and accounting (nonrepudiation). The security mechanism to protect the nonrepudiation is cryptographic in terms of a digital signature. However, the digital signature needs key management such as Public Key Infrastructure (PKI), which is a centralized system. For this paper, we propose a decentralizing paradigm or distributed system to prevent the nonrepudiation problem, whereas blockchain technology is a distributed system that guarantees security objectives in terms of accounting with the cryptographic wallet address. We conclude that blockchain technology is a solution for our use case.

*2.3. Ethereum Network*

Ethereum [14] is a distributed computing platform composed of a network of computers operating in a decentralized, self-governing, and democratic manner. Thus, Ethereum is gaining traction in a wide variety of applications.

The most attractive and promising concept in blockchain technology is the smart contract. The self-enforcing and event-driven characteristics of Ethereum allow some online actions to be conducted without a trusted third party [20]. There is a substantial community and a platform for the exchange of information. A smart contract encrypts the rules and obligations. An address, states, and functions make up a smart contract. A smart contract address is used to identify a contract implemented on the Ethereum Blockchain Network [21].

Ethereum performs smart contracts and launches distributed application. The front end may be deployed as a web application with a solidity smart contract serving as the back end.

*2.4. Transaction Fee*

Smart contracts and data are handled in the blockchain network through a transaction process. The smart contract data size and complexity determine the transaction fee [22]. In addition, the sender of the transaction must specify a charge. The node confirms the

transaction by obtaining a fee. Setting reasonable fees is essential to ensure that transactions are handled quickly.

The essential Ethereum–user interaction is a financial transaction or a smart contract. The user inserts a value, a gas, and a gas price, and Ethereum will execute it via Virtual Machine (EVM) until it is completed or runs out of gas. At this point, it joins the transaction pool: a waiting queue for transactions that historically presents itself with many backlogs until a miner decides to mine it. Then, it joins the blockchain itself [13].

One thing that makes Ethereum stand out is that it has an internal metering variable called "gas" that measures how complicated each transaction on the blockchain is. Each transaction has an execution cost set by an algorithm that is measured in gas units. The more complicated a transaction is, the more gas a user must have on hand and pay for the transaction to be recorded and carried out [23].

Ethereum Gas is the unit used to measure the computational costs of running a smart contract, which is paid for by the user. The gas mechanism gives the reward in Ether to the miner, who makes the transaction happen [21].

The gas mechanism is explained as follows:

1. Users define *gasLimit* and *gasPrice* in creating transactions. The *gasPrice* is the cost for each step of the execution, and *gasLimit* is the maximum gas in the transaction.
2. Ethereum calls the *feeMax (gasPrice x gasLimit)* from a sender and broadcasts an execution required to all nodes when a transaction is sent. The transaction will be waiting in the transaction pool.
3. The miner node gets the transaction from the transaction pool and runs it. The node prioritizes the requirements by *feeMax* and gets the highest reward.
4. The *gasUsed* is computed and accumulated at each stage of processing. Simultaneously, the EVM verifies that *gasUsed* is less than *feeMax*. If this is the case, execution continues until completion, and the *excess gas (feeMax–gasUsed)* is returned to the sender. If not, the execution is stopped, and a *gasLimit* exceeded exception is thrown.

## 2.5. Related Work

The main contributions of this work are to protect the security issues in traceability, nonrepudiation, and reliability with the dementalized paradigm. Thus, this work proposes the framework and smart contract for the smart logistics business with the home delivery use case. Moreover, the performance in terms of transaction fee and response time are discussed. In Section 1, we compared the related work of blockchain technology in supply chain and logistics. Most previous works proposed the study in the management dimension. Table 1 shows the key features of the research in the implementation of blockchain technology, with the supply chain and logistics described below:

- Business: Comparing business-to-business (B2B) and business-to-customer (B2C). The characteristics of each business are different: the size of the business, the complexity of the process, or the requirements. Home delivery, our use case, is business-to-customer.
- Requirement: The key requirements of smart logistics are traceability and reliability.
- Security: These are the objectives the framework needs to protect, such as confidentiality, integrity, and nonrepudiation.
- Approach: The security mechanism and other techniques that support the main requirements of smart logistics. Some previous studies have applied the Internet of Things and machine learning to increase reliability.
- Platform: The blockchain network that is the private blockchain (Hyperledger Fabric) and public blockchain (Ethereum).
- Performance: The performance metrics. Some related studies are evaluated in terms of the function and quality of the product. Most previous studies discuss the response time and latency. However, the transaction fee is an important issue that needs to be discussed.

**Table 1.** Related work on the application of blockchain in smart logistics.

| Key Feature | | [7] | [10] | [24] | [25] | [26] | Our Approach |
|---|---|---|---|---|---|---|---|
| | **Business** | B2B | B2B | B2C | B2B | B2B | B2C |
| Requirement | Reliability | ✓ | ✓ | ✓ | | | ✓ |
| | Traceability | ✓ | ✓ | ✓ | ✓ | ✓ | ✓ |
| Security | Confidentiality | ✓ | ✓ | ✓ | | | ✓ |
| | Integrity | ✓ | ✓ | ✓ | | | ✓ |
| | Nonrepudiation | ✓ | ✓ | ✓ | | | ✓ |
| Approach | Private Blockchain | ✓ | ✓ | | | | |
| | Public Blockchain | | | ✓ | | | ✓ |
| | IoT Based | ✓ | ✓ | ✓ | ✓ | | |
| | AI/Machine Leaning | | | ✓ | | | |
| Platform | Ethereum | | | ✓ | | | ✓ |
| | Hyperledger Fabric | ✓ | ✓ | | | | |
| | N/A | | | | ✓ | ✓ | |
| Performance | Functionality | | ✓ | | | ✓ | |
| | Transaction Fee | | | | | | ✓ |
| | Time | ✓ | | ✓ | | ✓ | ✓ |
| | Quality of Product | ✓ | | | ✓ | | |

Table 1 summarizes all the previous studies that applied blockchain technology to fulfill the smart logistics requirement and security requirement, especially the traceability. However, some studies were implemented on the private blockchain named Hyperledger Fabric. This approach suits a closed system. On the other hand, the main contribution of this study requires a public system. Thus, the Ethereum network or public platform is our selection for this research. Moreover, the transaction fee (gas used) and response time are the main points to note. Most previous studies discuss time issues but do not discuss transaction fees in the Ethereum network.

This paper focuses on the framework and smart contract design and implementation of the open system (Ethereum). Thus, the transaction fee and response time need to be the focus, whereas the functional requirements and security requirement still serve the home delivery business.

## 3. The Home Delivery Use Case and Its Security Issues

### 3.1. The Home Delivery Use Case

This research starts with a famous food delivery company in Thailand as an example of home delivery use case. The food delivery processes are published via the official website. Moreover, we interview the customer, shop or restaurant, and rider to confirm the process.

From the preliminary study of a food ordering platform, three stakeholders and one platform exist. The stakeholders are the client or customer, the restaurant the producer, the food delivery driver (rider) and the food delivery platform the distributor.

Each stakeholder accessed the system through a mobile application, while the foot delivery platform was a black box system. The platform is a centralized system that communicates through the HTTPS protocol and authenticates via email, Facebook, or phone. The payment methods are cash and credit card. Figure 1 illustrates the process.

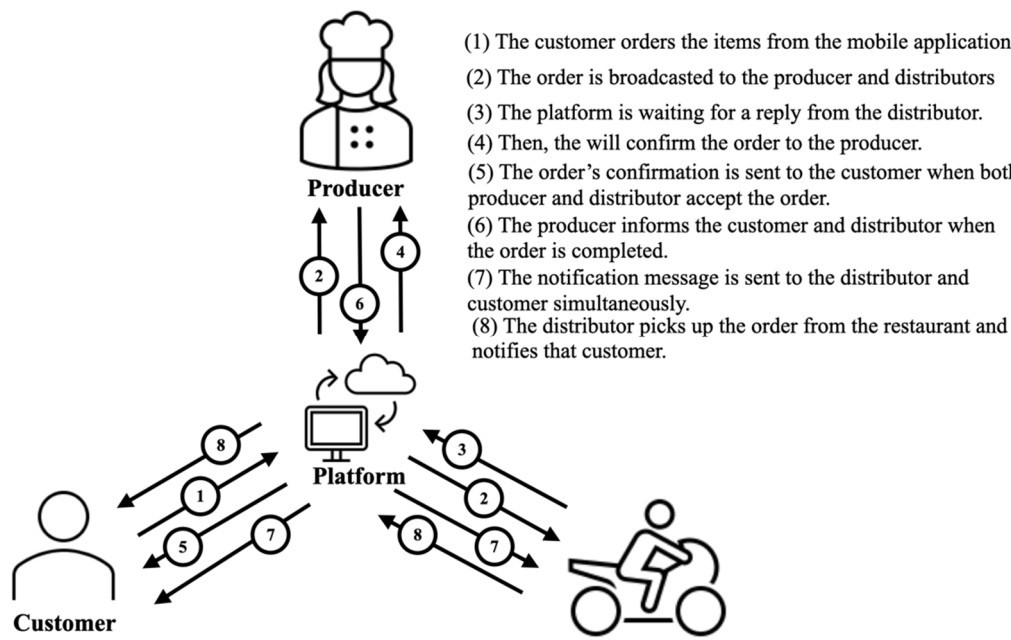

(1) The customer orders the items from the mobile application.

(2) The order is broadcasted to the producer and distributors

(3) The platform is waiting for a reply from the distributor.

(4) Then, the will confirm the order to the producer.

(5) The order's confirmation is sent to the customer when both producer and distributor accept the order.

(6) The producer informs the customer and distributor when the order is completed.

(7) The notification message is sent to the distributor and customer simultaneously.

(8) The distributor picks up the order from the restaurant and notifies that customer.

**Figure 1.** The workflow of a food delivery platform is the use case of this research. The producer is a restaurant, while the customer is a client. The distributor for the food delivery normally is a rider.

Figure 1 shows that (1) the clients select items from the item list via the mobile application. When they check out, they must choose a payment method: cash or credit card. The credit will be retained in the platform's payment system if they choose a credit card. On the other hand, clients must prepare the cash to pay riders directly when choosing the cash payment method. (2) After clients confirm the order, it is simultaneously forwarded to the restaurant and rider. The rider decides to accept or reject the incoming order. They must wait for a replacement if they reject it and cannot return to the previous order. Additionally, they must compete with other riders to accept the order. (3) On the restaurant's part, they need to wait for the rider's confirmation and double-check the ingredients for each item before they decide to accept or reject the order. The rider will receive a rejection notice if the restaurant rejects this order. The platform will reimburse the customer's credit. (4) When the restaurant receives the order, they begin cooking. They adjust the stage of the order via the mobile application. The restaurant notifies the rider via the application when the order has been completed. (5) The rider picks up the order. The riders must advance the amount if the payment is paid in cash. Then, the rider delivers the order that was placed by a mobile application. (6) The rider contacts the customer via the application and directly sends items to the customer. The consumer pays the rider if there is a cash payment. Then, the rider terminates the system via the application if the delivery process is complete. After that, the restaurant receives the money through the system. (7) If the rider does not contact the client, the rider must notify the platform and engage the verification procedure to solve the customer no-show issue or any problems. The money will be reimbursed to the rider or restaurant, while the platform absorbs all transaction costs.

### 3.2. The Security Flaws

The food delivery industry has existed for a long time and has faced various challenges. Many food delivery scams have emerged and been reported in the news and on social networks, such as the customer receiving the wrong items, the customer not being at the food delivery point (a no-show), or the product not being delivered directly to the customer. In addition, riders can steal products that have been paid for with a credit card. Paying cash causes a disclaimer problem, affecting customers, shops, and riders. Each step of the food

ordering process can experience flaws that could lead to fraud. From the investigations, we analyze the fraud problems as follows:

- Fake location: The customer sets a fake location to reduce the service charge because the service charge depends on the distance between the customer and the shop. If the customer then changes their location to the other point, it increases the rider's costs. The platform should set a new service charge policy for real-time service charge calculation.
- Mistaken items: When the customer receives the product, it is found that there is a mistake in the number or type of food items. The issue has several vulnerabilities, such as the restaurant arranging food that does not match the items that have been ordered. If they change some options, the customer should be informed directly or through the system. The shipper should verify the quantity and food items before shipping. This process ensures transparency between the store and the shipper.
- Fake orders: The no-show of the recipient may have been caused by a fake order. The system allows the order to be for another recipient. This problem often occurs with cash-on-delivery orders. All responsibility falls on delivery because the sender has paid in advance. In other cases, the employee cannot contact the recipient, resulting in the inability to send the item.
- Lost package: Some buildings have a mechanism that allows items to be placed in a common area to allow for social distancing, resulting in the loss of packages because the recipient did not receive the package, or the package was not actually delivered to the recipient. The status in the system states that the delivery is complete. However, the items did not actually reach the recipient due to this phenomenon.
- Fraud fee: The system is centralized and closed to the platform owner. The store and delivery staff are not authorized to access or inspect the contents. Only the owner of the platform can manage the data. The transport staff and the restaurant were skeptical of the controversies in the news and the social networks in Thailand.

From the issues stated above, it can be determined that most issues are due to untraceable mistakes. The primary reason for this is that the transaction is also cash-based, which raises the chances of fraud. When ordering food with a credit card, this problem is eliminated. The restaurant checks the quantity and kind of food. Thus, the rider double-checks the order. They take a photo of the items as evidence.

This method also applies to challenges on the recipient's side. If the recipient does not receive the shipment or an item is missing, there is no way to confirm that the item was lost during delivery. Thus, the platform should be a system that is reliable, transparent and involves nonrepudiation, including a cashless platform to prevent the problems mentioned above.

## 4. The Proposed Framework

This research presents a home delivery system using the Ethereum smart contract. The system's main objective is to enhance the security of logistics services to address the nonrepudiation issue. The transactions are stored in the blockchain network, while application data are stored in the application's database. Stakeholders manage each transaction: customer, producer, and distributor. Moreover, it is transparently processed in the blockchain network. All stakeholders are authorized to edit based on their privilege.

When the conflict occurs, the transactions in the blockchain will be the evidence. Through this approach, stakeholders are assured that all transactions are secure, traceable, and reliable because of the dominant characteristics of the blockchain. The framework details are described in the following sections.

### 4.1. Secured Home Delivery Model

We assume that the secure home delivery service has stakeholders that exchange information over the secured channel with a timestamp for each smart contract method.

The transactions are the order list stored in the blockchain network. The order of the home delivery model is shown below:

**Definition 1.** *Stakeholder: A stakeholder, $\mathcal{R}$, is a person or company concerned with the home delivery process and involved in* an *order. Let $\Psi$ represent the set of all wallets in the proposed system and $\mathcal{R} = \{\alpha, \beta, \chi\}$ represent any producers, customers, and distributors. Let $\sigma \in \mathcal{R}$, then*

$$\sigma = \langle i_\sigma, \omega_\sigma, \text{desc}, \text{roles}\rangle \tag{1}$$

*where:*

- *$i_\sigma$ is a unique and readable identifier for $\sigma$.*
- *$\omega_\sigma \in \Psi$ is a unique wallet represented by 40 hexadecimals.*
- *desc is a stakeholder's information, represented in textual or binary encoding with varying data length.*
- *roles ($\mathcal{R} = \{\alpha, \beta, \chi\}$) is any person in the stakeholder set. For the food delivery platform, the stakeholders are restaurant ($\alpha$) as a producer, customer ($\beta$), and rider ($\chi$) as a distributor.*

The primitive data of stakeholders are stored in an external system such as a third-party database. The information of each stakeholder is different. It is presented in textual, JSON, or binary encoding, depending on the core platform.

**Definition 2.** *Smart Contract's Method (SCM): A smart contract's method, ($\varsigma$), is the list of sequential methods implemented in a smart contract that is executed in the blockchain network and consumes the transaction fee. A smart contract manages the order, updates the status, and searches. The SCM is shown in Figure 2 and described in Table 2.*

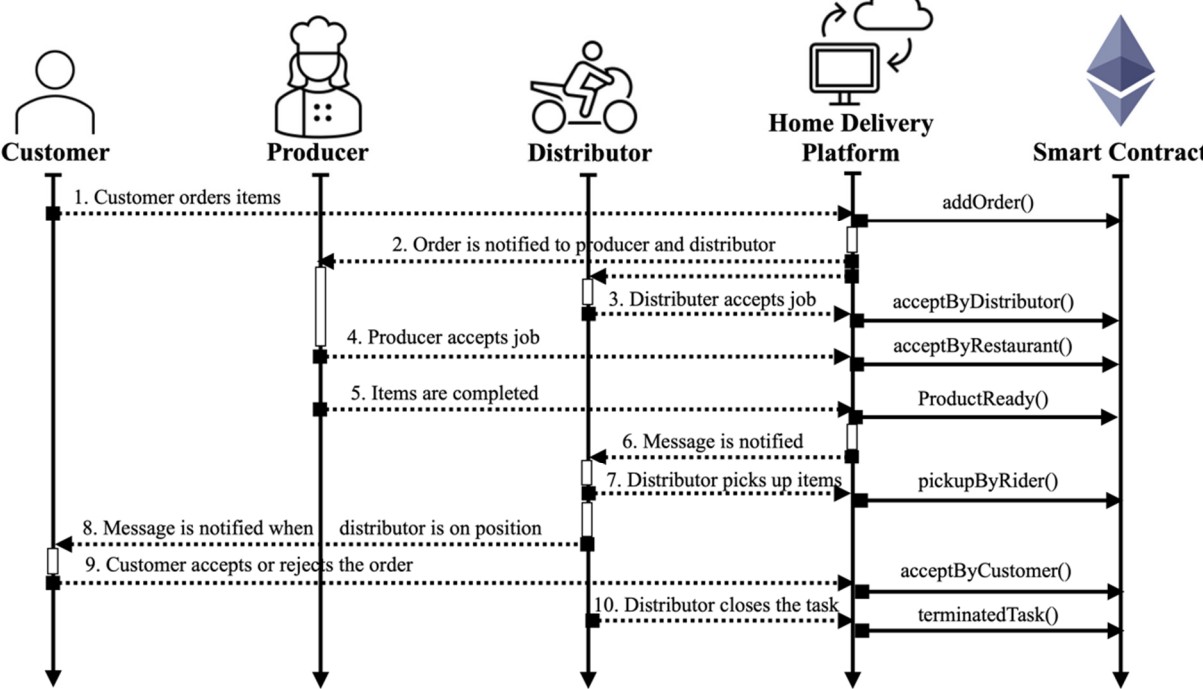

**Figure 2.** The workflows of the proposed architecture. The dotted line represents the traditional operation of the home delivery platform, while the solid line represents the methods in the smart contract. This diagram shows the write permission method only because they consume *gasUsed*.

**Table 2.** The smart contract method to secure a home delivery model.

| Method/Short Name | Parameters | Caller | Description |
|---|---|---|---|
| *addOrder()/ao* | $\omega_\alpha$, $\omega_\beta$, *desc*, $t_{ao}$ | Customer | This method adds the transaction to the blockchain network. All parameters are initiated with 0 except $\omega_\alpha$, $\omega_\beta$, *desc*, and $t_{ao}$ as the parameters. The transaction will be created in a secured block and linked with the other blocks in the blockchain. |
| *acceptByDistibutor()/ad,* | $\omega_\chi$, $t_{ad}$ | Distributor | When the distributor receives the new notification message from the application, the distributor calls the function to accept this task. The $\omega_\chi$, $t_{ad}$, and *status* are updated in the smart contract. |
| *acceptByProducer()/ap* | $t_{ab}$ | Producer | The producer accepts the order when there is a distributor accepting the order. The timestamp, $t_{ap}$, and *status* of the transaction will be updated. |
| *ItemsReady()/ir* | $t_{ir}$ | Producer | This method sends a message to the distributor that the items are ready to pick up. This function updates the value of $t_{ir}$. |
| *pickupByDistributor()/pd* | $t_{pd}$ | Distributor | After the distributor checks the order is correct, s/he picks up the product and confirms the order. The $t_{pd}$ is updated as evidence to guarantee the order. |
| *acceptByCustomer()/ac* | $t_{ac}$ | Customer | When the customer has checked and accepted the order, this method sets the $t_{od}$ and changes the status to accept. |
| *closeOrder()/co* | $t_{co}$ | Distributor | The distributor is the last person who closes the order when all operations are completed. The $t_{cd}$ is stamped to log the transaction. |

**Definition 3.** *Order List: An order list, $\Theta$, is the item list stored in the smart contract. An order, o $\in \Theta$, is defined as a 3-tuple,*

$$o = <\Psi_o, \tau, \text{desc}, \text{status}> \tag{2}$$

*where:*

-   *$\Psi_o$ is the set of three stakeholders' wallets involved in an order. Let $\Psi_o = \{\omega_\alpha, \omega_\beta, \omega_\chi\}$ represent the stakeholders of the order o. Thus, $\omega_\alpha$, $\omega_\beta$ and $\omega_\chi$ represent the wallet of the producer, customer, and distributor, respectively.*
-   *$\tau$ is a set of timestamps indicating the response time and order list for each method in SCM. $\tau = \{t_m, m \in \varsigma\}$.*
-   *desc is the description of the product or parcel and its status, such as the menu and drink items for the food delivery system, or the details of the parcel.*
-   *status indicates whether an order was successful or what state it is in.*

*4.2. System Architecture*

The home delivery stakeholders consist of three categories: customer, producer, and distributor. All participants connect to the system via the home delivery platform, as illustrated in Figure 3. In contrast, the smart contract is the system's main component that enforces a policy of home delivery service. All stakeholders need to follow up. The policy is represented as a contract and stored in the blockchain network. As a result, the policy does not change after the system begins operation.

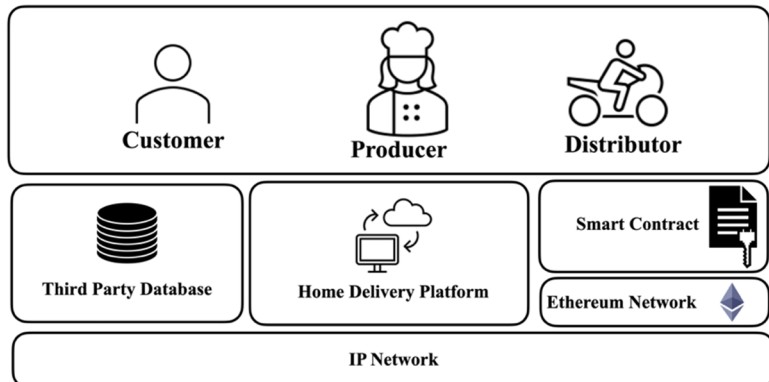

**Figure 3.** The proposed architecture enhances security by adding the smart contract and the Ethereum Network. All services are communicated over the secured IP network.

The customers, producers, and distributors are the Ethereum blockchain users. The customer starts the transaction. The producer and distributor update and terminate the status depending on the situation. Each stakeholder can retrieve or update data on the transaction. This research demonstrates our framework for food delivery. Thus, each role includes different actions that the stakeholder can play in the smart contract, as follows:

- Customer: The customer starts the transaction by selecting the order retrieved from the home delivery platform. After that, they confirm the order and specify the payment method (cash or credit card). Finally, the customer confirms the order when they meet and receive the product from the distributor.
- Producer or Restaurant: The role of the restaurant is to accept the order and wait for the rider to accept the order. On the other hand, the restaurant can reject the order if they cannot cook the order to meet the contract. Moreover, they need to confirm the order and submit the product to the rider when it is ready.
- Distributor: Firstly, the rider accepts the order. Then, they pick up the product and send it to the customer. In addition, the rider terminates the transaction when the product is delivered to the customer.

From a technological perspective, all stakeholders access our approach via the home delivery platform, a mobile/web-based/native application, depending on the business. The platform access database is via the application programming interface (API). Moreover, the database may be the relational database management system (RDBMS) or nonrelational database (NoSQL). On the other hand, the smart contract is in the blockchain network. For this research, we designed it based on the Ethereum network. The smart contract was implemented with Solidity programming language. All technology components are communicated via an IP network or the Internet.

*4.3. Software Functions*

The home delivery architecture comprises two parts: distributed application and blockchain network. The distributed application is the user interface implemented with mobile or web-based applications. All users exchange data over a secure protocol. On the other hand, the blockchain network is the connection between distributed applications and the smart contract via the blockchain network. The workflow of the proposed architecture is shown in Figure 2.

All stakeholders access the application, like with the previous application shown in Figure 1. However, this research replaces some functions with the smart contract. The primitive data of the application are still stored in a third-party database, depending on the platform. The transaction and its statuses will be stored in the blockchain network and manipulated by function in the smart contract. The functions are divided into two groups: *addOrder()* as shown in Algorithm 1 and *setStatusAndTimeStamp()* as shown in Algorithm 2. A list of functions and descriptions are given in Table 2.

Algorithm 1 is used for new order adding. The parameters are the wallet address of the customer and producer and the order description, represented in JSON format. The parameters are initiated in this algorithm. We assign a zero value to the wallet address of the distributor, the timestamp in $\tau$, and the order's status. The authorized wallet address can only add new orders to the blockchain network.

---

**Algorithm 1** Add new order to blockchain

---

**Let** $\Psi$ and $\Psi_o$ be the wallet address of the stakeholder
$\quad$ $\tau$ is the set of state's timestamps
$\quad$ $\Theta$ is the order list in blockchain
**Input**:
$\quad$ $\omega_\alpha$, $\omega_\beta$, *desc*
**Step**:
$\quad$ $\Psi_o \leftarrow \varnothing$
$\quad$ $\omega_\chi$, *status = 0x0*
$\quad$ $\forall_{t \in \tau} [\tau(t) \leftarrow 0]$
$\quad$ **if** $\omega_\alpha \notin \Psi \vee \omega_\beta \notin \Psi$ **then**
$\quad$ return
$\quad$ $\Psi_o = \Psi_o \cup \{\omega_\alpha, \omega_\beta, \omega_\chi\}$
$\quad$ **push** $<\Psi_o, \tau, desc, status>$ **to** $\Theta$

---

The function creates a new block containing all participants' wallet addresses, the order description and status, and all initial timestamps. Finally, the smart contract links the new block to the blockchain.

---

**Algorithm 2.** Update status and timestamp by ID

---

**Let** $\Psi$ be the wallet address of the stakeholder
$\quad$ $\tau$ is the set of state's timestamps
$\quad$ $\varsigma$ is the set of contract methods
**Input**:
$\quad$ *id*: the order id
$\quad$ $m \in \varsigma$
$\quad$ $\omega \in \Psi$ is any stakeholder
$\quad$ *newStatus*
**Step**:
$\quad$ $<\Psi_o, \tau, desc, status> = \textbf{\textit{findOrderByID}}(id)$
$\quad$ **if** $\omega \notin \Psi \wedge \omega \notin \Psi_o$ **then**
$\quad$ return
$\quad$ *status = status | newStatus*
$\quad$ $\tau(m) = getTimestamp()$

---

The main functions of Algorithm 2 are to update the order's status and the timestamp of each process. The wallet address of the caller authorizes this function to search by order ID. The operation will be rejected when the caller is unauthorized. The new status will be set to the order description, and the blockchain network time is the timestamp.

## 5. The Experimental Setup

### 5.1. Testbed and Scenarios

The objective of this experiment is the performance evaluation of a smart contract in a closed environment. The relationship between the data length of order description and the performance metrics are represented by the transaction cost and application response time. This experiment involves black-box testing. We evaluated the overall response time and specified the transaction cost of each function.

We tested the application on MS Windows 11 installed on a laptop: 11th Gen Intel® Core™ i5 2.70 GHz, 16 GB RAM, and 256 GB SSD. We selected the Ethereum network with POW as a consensus algorithm on Ganache 2.5.4. This experiment did not connect to an

external computer to eliminate the effect of network delay. The smart contract was implemented with Solidity language and compiled by Remix IDE (https://remix.ethereum.org (accessed on 15 August 2022)). In addition, we implemented the application with NodeJS, which connected the blockchain network via Web3.js.

The scenario comprised 20 restaurants, 20 customers, and 20 riders. The 100 transactions with random data length were randomly pushed to the blockchain network via the script. This simulation does not simulate the time taken up by cooking, riding, and the restaurant and rider deciding to accept the order.

The dependent variable was data length, while the independent variables were the transaction cost and response time. The control variables were tested in a closed system to reduce the effect of network delay.

### 5.2. Performance Metrics

The objective of this experiment was to evaluate the cost and response time for a transaction. The *gasPrice* used in the experiment was 3 GWei, as suggested by the Metamask wallet [21]. The performance metrics were as follows:

- *Transaction cost*: The transaction fee or *gasUsed* was the cost of a blockchain transaction. Each function consumes a different cost depending on the coding style and data length. The actual cost was the *gasPrice* multiplied by the *gasUsed*. For this experiment, the *gasPrice* was 3 GWei. Thus, the actual cost was 3 GWei multiplied by the *gasUsed* in Table 3.

- *Response Time (RT)*: We focused on the response time of each function. The overall response time was the sum of latency in the calling function ($t_f$) and the processing time in the smart contract method. Algorithms 1 and 2 consist of three parts: *time_of_validate* ($t_v$), *time_of_set* ($t_s$), *time_of_add* ($t_a$), and *other_time* ($t_o$). The *time_of_validate* is in the parameter's validation state. Then, the *status* or any parameters are updated in the *time_of_set* state and closed with the *time_of_add* state that updates the new transaction in the *Order List*. Moreover, the *other_time* is the additional time when the smart contract method needs to execute the helping method. Hence,

$$RT = \sum_{t_i \in \tau} \left( t_{if} + t_{iv} + t_{is} + t_{ia} + t_{io} \right).$$

**Table 3.** Average gas used and response time for each function.

| Function | Gas Used (Unit) | | Response Time (ms) | |
|---|---|---|---|---|
| | Average | s.d. | Average | s.d. |
| *addOrder()* | 760,081.47 | 235,824.27 | 2431.66 | 759.93 |
| *acceptByProducer()* | 194,401.32 | 73,442.89 | 406.97 | 44.40 |
| *acceptByDistributor()* | 194,687.32 | 73,442.89 | 387.42 | 51.63 |
| *ItemReady()* | 173,463.32 | 73,442.89 | 319.71 | 47.85 |
| *pickupByDistributor()* | 173,486.32 | 73,442.89 | 325.93 | 45.14 |
| *acceptByCustomer()* | 173,487.32 | 73,442.89 | 381.32 | 45.55 |
| *closeOrder()* | 194,733.32 | 73,442.89 | 328.98 | 60.25 |
| Overall | 1,864,340.41 | 447,259.27 | 4582.00 | 758.43 |

## 6. Results and Discussion

### 6.1. Performance Issues: Gas Used

The dependent variable was the data length of order description. The range of data length was 187–1133 bytes (mean is $630 \pm 261$), whereas the independent variables were transaction cost and response time, discussed in the following:

Table 3 shows the *gasUsed* for each function and overall. The *gasPrice* was 3 GWei. Thus, the average transaction cost was $1,864,340.41 \times 3$ GWei, which is $5,593,021.22$ GWei (1 GWei = $10^{-9}$ Ether). The Ether value at the time of writing is 1026.89 USD, resulting in the overall transaction cost in fiat being $5,593,021.22 \times 10^{-9} \times 1026.89$, or 5.74 USD.

The data length directly affects the hash and block process of the blockchain. The *addOrder()* function consumes the maximum gas used because it needs to process the order description in the blockchain network (the statement "push < $\Psi_o$, $\tau$, *desc*, *status*> to $\Theta$" in Algorithm 1).

Figure 4 shows the relationship between data length and gas used in the *addOrder()* function. This trend line shows that the gas used is the direct variant, with the data length as a linear function. The trend line is $y = 899.3x + 193,200$ where $x$ is the data length, $y$ is the gas used, and $R^2 = 0.9940$.

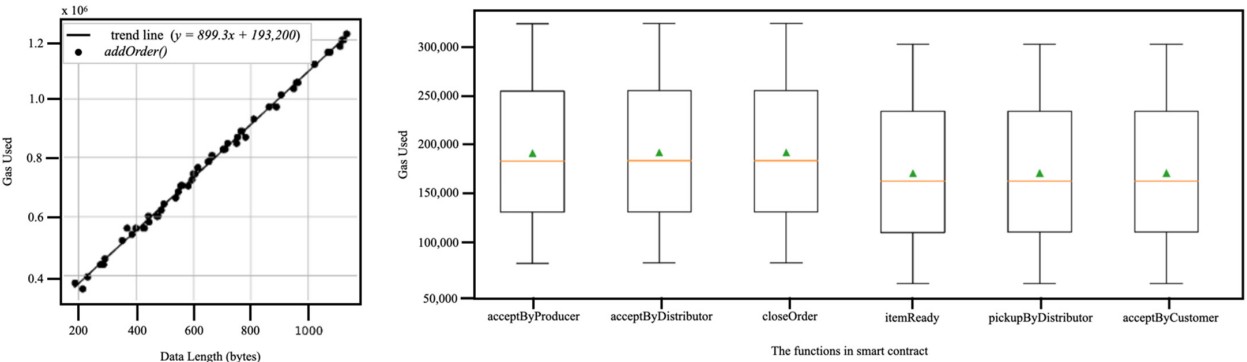

**Figure 4.** The gas used of *addOrder()* increases as a linear function ($y = 899.3x + 193,200$, $R^2 = 0.9940$). The gas used by other functions is not significantly different (F-score = 34.47, F-critical = 2.24, *p*-value = 0.00).

On the other hand, we compare the average gas used of six functions in the *setStatusAndTimeStamp()* group with the ANOVA method. The results show that the F-score = 34.47, F-critical = 2.24, and *p*-value = 0.00. Thus, we conclude that the average gas used for each function is not significantly different. As shown in Algorithm 2, the instruction is compo of three groups: (1) sequential search order by ID, (2) wallet address validation, and (3) status and timestamp updating. The sequential search is view mode in the smasedrt contract. This does not affect the gas used but affects the response time. In contrast, the gas used for the validating and updating instruction is at the same level.

*6.2. Performance Issues: Response Time*

Table 3 shows the response time. The response time of the *addOrder()* function takes longer than another function by 600% because this function encrypts the longest data length compared to other functions. The function will take longer to encode as the data length increases. The overall response time will be longer, as shown in Figure 5. This trend line shows that the response time is the direct variant, with the data length as a linear function. The trend line is $y = 2.84x + 638.87$, $R^2 = 0.9802$ where $x$ is the data length and $y$ are the response time.

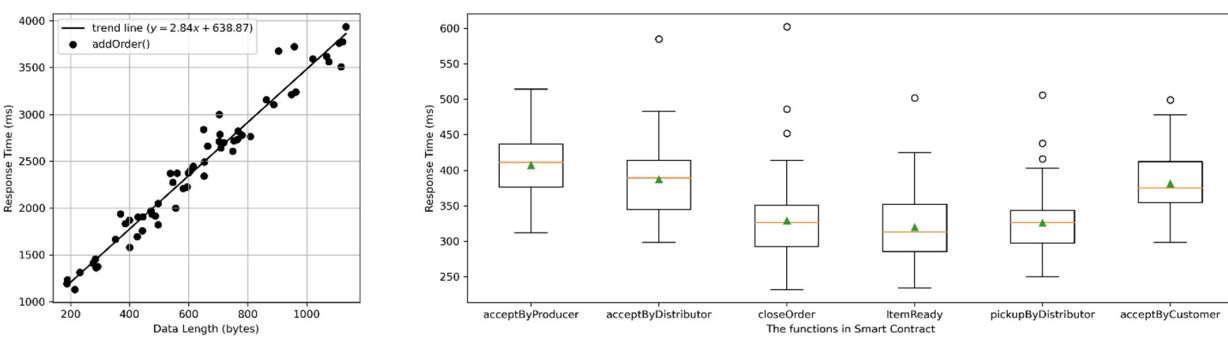

**Figure 5.** The response time of *addOrder()* increases as a linear function ($y = 2.84x + 638.87$, $R^2 = 0.9802$). Moreover, there is no significant correlation between the data length and response time.

For the functions in the *setStatusAndTimeStamp()* group, the response times are clearly different. There is no significant correlation between the data length and response time. The main statements are updating the timestamp and the status. Thus, each function's hash time and block time are not different. On the other hand, the operation and data transmission between processes cause latency and result in the transaction's overall response time not being predicted correctly.

### 6.3. Security Issues and Policy

The main objective of this research is to prevent the nonrepudiation problem. Blockchain technology is a tool to control and monitor the secured process only. Traceability is an essential characteristic of this technology. However, as explained in the previous section, it cannot prevent all security flaws. The policy needs to control and incorporate the blockchain's features to increase the completeness of security prevention. We introduce the security policy that is controlled by the proposed framework as follows:

- The mistaken item problem is eliminated because the producer declares the items to the distributor. Then, the distributor confirms the type and number of items via our framework and notifies the customer as an alert message. This includes the details of the order to double-check the correctness.
- The package loss problem is prevented because the distributor confirms the item's correctness in the face of the producer. In addition, the customer verifies the items after receiving the order and confirming it with the distributor. The additional technique is taking a photo as evidence in case an item is lost. The IPFS technology is a tool for securing media files.
- The problem of fake orders or locations will be eliminated when cashless payment is used in this service. Payment with a credit card guarantees that fake order and location events are prevented. Moreover, blockchain technology introduces the coins or coupons of the business. The service creates a secured coin or coupon to reduce the cash for the promotion or campaign. Moreover, the coin will support the transaction fee payment between the producer, distributor, and platform. Moreover, the cashless method will reduce the fraud fee because nobody touches money in the system. In addition, all the transactions are recorded and monitored by the miner nodes in the blockchain network.

### 6.4. Limitations

We proposed, demonstrated, and evaluated a conceptual framework based on blockchain technology. We demonstrated our framework on the local blockchain network, implemented the mobile application, and evaluated the application on the emulator. The effects of network speed and hardware resources were not evaluated. Thus, the experimental results cannot be used as a reference in real-life applications.

The limitation of our experiment is that we execute all test cases on the blockchain simulation named "*Ganache*". Testing on a real blockchain network can only evaluate the gas used. We cannot estimate the response time because the network bandwidth affects the experiment more than the response time of operation in the blockchain network. To reduce the response time, we need to increase the *gasPrice* to more than 3 GWei. On the other hand, this causes an increase in the overall transaction fees.

### 6.5. Open Issues

The transaction fee is too high in the preliminary application. Thus, transaction fee minimization is an important issue to consider in the future. This research points out that data length affects transaction fees and response times. Furthermore, the increase in gas used results in a decrease in the response time. Moreover, the opcode that is compiled from the smart contract will consume the gas. Each opcode consumes a different level of gas used. Thus, smart contract optimization is a solution for reducing the transaction fees [27–31].

In addition, the long data length that is processed in blockchain makes the gas used higher than a short data length. Thus, we introduce a data compression algorithm or data structure design to minimize the data length [32,33].

On the other hand, adding any process before calling the smart contract method will increase the response time in terms of both client device response time and computation power. As a result, the high response time may affect the user's satisfaction; every technique has unwanted side effects. Therefore, the research needs to strike a balance between the response time and transaction fee.

## 7. Conclusions

This paper demonstrates the application of blockchain technology to intelligent business-to-customer logistics: a home delivery framework. The main objective is to improve the nonrepudiation of transactions on home delivery services. The transparency of the system is improved by utilizing blockchain technology. All the actions in the system can be authenticated by the unique ID and controlled with the smart contract on the blockchain network. Authorized users can access and trace an order in case there is any conflict. However, our approach involves the addition of security issues. All the data from this system are stored in a centralized database or depend on the product owner. All the transactions are stored in the Ethereum network to guarantee nonrepudiation and evaluated in terms of transaction cost and response time.

**Author Contributions:** Conceptualization, T.C. and C.J.; methodology, T.C. and C.J.; software, T.C.; validation, C.J.; formal analysis, C.J.; investigation, T.C.; resources, T.C. and C.J.; data curation, T.C.; writing—original draft preparation, C.J.; writing—review and editing, C.J.; visualization, C.J.; supervision, C.J.; project administration, C.J.; funding acquisition, C.J. All authors have read and agreed to the published version of the manuscript.

**Funding:** This research work was financially supported by Walailak University Personal Research Fund, contract number WU65208.

**Institutional Review Board Statement:** Not applicable.

**Informed Consent Statement:** Not applicable.

**Data Availability Statement:** Not applicable.

**Conflicts of Interest:** The authors declare no conflict of interest.

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
