# Peer review of "Security Enhancement in Smart Logistics with Blockchain Technology: A Home Delivery Use Case"

_informatics, doi:10.3390/informatics9030070_

Round 1

Reviewer 1 Report

The article is biased and many years late. It does not bring any news. It is an example of another application that could use blockchain technology. In fact, for a large scale of simultaneous transactions and delays occurring in the network, the use of such a system would be unprofitable and ineffective. The authors indicated a set of problems that they would like to solve by using blockchain as an additional database system. Can the presented problem not be solved with a slightly more complex system without blockchain technology?

Reviewer 2 Report

1. The reasons of using blockchain technology at home delivery could be clarified specifically.

2. The time cost could be considered in proposed method and compared with the existing applications.

3. Adding the comparison table would be better for understanding the significance of this work.

4. It should be detailed at how to prevent the non-repudiation problem more specifically to strengthen the study.

5. It is a real application applied on the real life with the modern technology.

6. Some text writing should be amended.

Reviewer 3 Report

This paper presents a framework using blockchain for in-home delivery case. Overall, the paper is interesting and well-presented.

1.       Section 2.1, related work. The authors are expected to compare the similar work using blockchain to solve non-repudiation problem in home delivery in depth.

2.       Why do the authors use Ethereum as the blockchain platforms, compared to other platforms, such as Hyperledger fabric, Stellar, Hyperledger Sawtooth, R3 Corda…?

3.       The authors should compare the proposed work with related work in terms of performance.

4.       Figure 2 and 3 are blurry.

5.       Line 52, “[3]. exchanges between…”, not a sentence.

6.       There are some typos or grammar errors, such as industry 4,0 not 4.0. Please proofread the paper again.

7.       Line 108, “…The parcel delivery is an example” should be “Take parcel delivery as an example”.

8.       Line 139, “Each node peers at other random nodes”. Please revise this sentence. 

Round 2

Reviewer 3 Report

The authors have addressed all my comments.